# Genetic Structure and Genetic Diversity of the Endemic Korean Aucha Perch, *Coreoperca herzi* (Centropomidae), in Korea

**DOI:** 10.3390/ani13162614

**Published:** 2023-08-14

**Authors:** Kang-Rae Kim, Sang Ki Kim, Mu-Sung Sung, Jeong-Nam Yu

**Affiliations:** 1Animal & Plant Research Department, Nakdonggang National Institute of Biological Resources, Sangju 37242, Republic of Korea; kimkangrae9586@gmail.com (K.-R.K.); ivoice@nnibr.re.kr (S.K.K.); 2Muldeuli Research, Icheon 12607, Republic of Korea; rkrtlqndj@naver.com

**Keywords:** gene flow, reintroduction, genetic structure

## Abstract

**Simple Summary:**

*Coreoperca herzi* is a freshwater fish endemic to Korea. So far, studies on *C. herzi* have mainly focused on ecological studies, but studies on its genetic diversity are lacking. For the first time, we investigated genetic diversity and structure using mitochondrial DNA data in this study. The genetic diversity was found to be low. In addition, the translocated population, Yangyangnamdaecheon, was confirmed to have originated from the Han River water system population as a result of genetic similarity. This study provides information on the genetic diversity, genetic structure, and translocated populations of *C. herzi*, providing a basis for the conservation of the species.

**Abstract:**

The Korean endemic aucha perch, *Coreoperca herzi*, belongs to the family Centropomidae. Thus far, studies on *C. herzi* have focused on mitochondrial genomes, egg development, and early life history, while studies on their genetic diversity or genetic structure are lacking. We investigated these aspects in this study using mitochondrial DNA data. Haplotypes were divided into the Hangang River, Nakdonggang River, Geumgang River, and southwest region water system populations. A translocated population, the Yangyang Namdaechun Stream, was confirmed to have originated from the Hangang River water system population based on haplotype distribution and genetic structure results. The *F*_ST_ of the mitochondrial DNA indicated distinct genetic differentiation in the Hangang, Nakdonggang, Geumgang, and southwest regions. According to *COI* and analyses, the analysis of molecular variance revealed a higher variance in the four water system groups (98.41%) than in the southwest region water system versus the Hangang River water system (80.27%) groups. This study presents basic data for conservation by providing extensive information on the genetic diversity, genetic structure, and translocation population of *C. herzi*.

## 1. Introduction

The Korean aucha perch, *Coreoperca herzi* Herzenstein (1896), is endemic to Korea and belongs to the order Perciformes and the family Centropomidae [1]. It is distributed throughout Korea and mainly inhabits places with fast currents and clean water [1]. *C. herzi* is a crucial species for domestic aquaculture, and its seed production has also been studied [2]. However, despite its commercial value, populations in the wild are declining due to habitat destruction, water pollution, and overfishing as human activities expand [3].

*C. herzi* is a commercial fish of major value, and wild broodstock is used for aquaculture. However, the genetic diversity of the offspring is low due to the small number of mothers. This low genetic diversity in offspring can have negative effects during interbreeding with natural populations [3,4]. Hence, understanding the extent and patterns of genetic diversity is essential for both natural and cultured populations [5]. These progeny populations with low genetic diversity are unsuitable as aquaculture resources or as sustainable aquaculture production resources. Therefore, evaluating the genetic diversity and genetic structure of the Korean wild populations is necessary to establish effective breeding and conservation strategies for wild populations. Studies on *C. herzi* have been conducted on the mitochondrial genome, egg development, and early life history, but there have been no studies on genetic diversity or genetic structure [6,7,8,9].

*C. herzi* are known to exhibit morphological differences in white spots between the Hangang River water system and other southwestern water systems, as well as variations in the morphological development of eggs [9]. Further phylogenetic studies are required to clarify these phylogenetic issues. In phylogenetic studies, mtDNA genes were reported to be useful for detecting inter-species and intra-species taxonomic divisions [10,11]. *C. herzi* have been manually translocated to the Yangyang Namdaecheon Stream (YYND) population, which belongs to the waters of Korea’s east coast [12,13]. Detecting the origins of translocated species is important for conserving native ecosystems, as these species can have detrimental effects on native species [14].

In this study, the genetic diversity and genetic structure of 19 *C. herzi* populations, which are endemic to Korea, were investigated using mitochondrial DNA (*COI*) datasets. This study may help in understanding the genetics and evolution of *C. herzi* and provide a basis for *C. herzi* conservation.

## 2. Materials and Methods

### 2.1. Sample Collection and DNA Extraction

*C. herzi* populations were sampled at 19 points in 2019 using fishing nets; detailed information is given in Figure 1 and Appendix A. *C. herzi* is endemic to Korea, and research on it does not require permission for animal ethics. The pectoral fin tissues of the fish samples were preserved in 99% ethanol. Genomic DNA was extracted from the preserved tissue of all specimens using the DNeasy Blood & Tissue kit (QIAGEN, Germantown, MD, USA) according to the manufacturer’s instructions.

### 2.2. MtDNA Sequencing

The primers for mtDNA (FishF1: TCAACCAACCACAAAGACATTGGCAC, FishR1: TAGACTTCTGGGTGGCCAAAGAATCA) were selected from Ward et al. [15], and the PCR was performed using a Mastercycler^®^ pro gene amplifier. For the PCR, AccuPower^®^ PCR Premix Kit (BIONEER Co., Daejeon, Republic of Korea) was used, and 1 μL of genomic DNA, 1 μL of each 1.0 μM forward and reverse primer, and 17 μL of tertiary distilled water were added and mixed to a final volume of 20 μL. The PCR conditions were as follows: pre-denaturation at 95 °C for 2 min, denaturation at 94 °C for 30 s, annealing at 52 °C for 30 s, and extension at 72 °C 35 times, followed by a final extension at 72 °C for 10 min, and termination at 4 °C. The *COI* gene PCR products were sequenced using an ABI 3730xl DNA Analyzer (Applied Biosystems, Waltham, MA, USA).

### 2.3. Statistical Analysis of Genetic Diversity in MtDNA

The mtDNA was examined via alignment using the ClustalW algorithm of the MEGA software ver. 11.0.1 [16] based on COI sequencing. The haplotype was extracted using the DnaSP software (Ver. 5.0 [17]). By using Network (Ver. 10.2.0.0 [18]) software to create a haplotype network, a median-joining network analysis was performed to read the affinity between genotypes. ARLEQUIN (Ver. 3.05 [19]) was used for genetic differentiation between groups, and an analysis of molecular variance (AMOVA) was performed. First, AMOVA was divided according to the distribution by water system (Southwest region water system vs. Hangang River water system; NNG, NDC, NSG, NDJ, NIM, NYS, MJJ, GYD, GND, SON, SSJ, YJS vs. HDC, HSC, HHC, HWS, HMW, HSSC, YYND). Second, AMOVA was divided according to the haplotype distribution (NNG, NDC, NSG, NDJ, NIM, NYS vs. GYD, GND vs. MJJ, SON, SSJ, YJS vs. HDC, HSC, HHC, HWS, HMW, HSSC, YYND).

## 3. Results

### 3.1. Genetic Diversity

The mtDNA sets showed genetic diversity among the 19 populations (Table 1). The number of haplotypes was 1–7, and the range of haplotype diversity was 0.00000–0.67857, with NSG, NDJ, NIM, HWS, YJS, and the YYND population showing the lowest diversity. The range of nucleotide diversity was 0.00000–0.00196, and it was highest in the HSC.

Nineteen populations were investigated using a 655-bp-long sequence in the *COI* gene region of the mtDNA. In total, 22 haplotypes and 84 polymorphic sites were identified. The H1 haplotype was found to be shared among the Hangang River watershed populations (HDC, HSC, HHC, HWS, HMW, and HSSC). H2–H12 haplotypes were shared only in the Hangang River water system population. H13–H15 were shared in the Nakdonggang River water system populations (NNG, NDC, NSG, NDJ, NIM, and NYS), whereas H16 was only present in the NYS. H17 was shared by the Geumgang River water system populations (GYN and GND), while H18 and H19 appeared only in the Geumgang River populations. H21 was shared by MJJ, SON, SSJ, and YJS, while H22 appeared only in the Seomjingang River population. The YYND population may have translocated from the Hangang River population by sharing the H1 haplotype with the Hangang River water system populations (Table 2).

### 3.2. Variations in the Genetic Differentiation of Populations

The *F*_ST_ values of mtDNA were −0.126–1.000, implying a high degree of differentiation. Except for NNG and NSG, which were highly differentiated due to non-shared haplotypes, the Hangang and Nakdonggang River water system populations had a genetic differentiation of 0.983–1.000. The Hangang and Geumgang River water system populations had a genetic differentiation of 0.979–1.000. The Seomjingang water system river populations (SON and SJJ) showed low genetic differentiation, with YJS at 0.069–0.273 and MJJ at 0.026–0.725.

### 3.3. Population Genetic Structure

In the median-joining network, haplotypes were divided into four groups centered on H1, H14, H17, and H21 (Figure 2). The first haplotype network demonstrated that all the populations in the Hangang River water system shared H1. The second haplotype group, H14, was shared by all groups except NSG, which shared H15 with NDC. A third haplotype group, H17, was shared by the Geumgang River water system populations, which had unique haplotypes, such as H18 and H19, as well. The fourth haplotype group, H21, was shared among MJJ, SON, SJJ, and YJS. The translocation population (YYND population) showed only the H1 haplotype, suggesting that it was translocated from the Hangang River water system.

We constructed an mtDNA-based phylogenetic tree of the 19 populations using the maximum likelihood (ML) method with PhyML (Ver. 3.0 [20]) (Figure 3). The phylogenetic tree was divided into four clades among the populations. According to the phylogenetic tree, the populations were assigned to the following groups: Clade (1) NNG, NDC, NSG, NDJ, NIM, NYS, and the YYND population; (2) GYD and GND; (3) HDC, HSC, HHC, HWS, HMW, and HSSC; and (4) MJJ, SON, SSJ, and YJS. Notably, the YYND population was grouped with Hangang River water system populations, indicating that it is an emigrant.

The AMOVA of *C. herzi* was performed based on the haplotype distribution (Table 3). The mtDNA was divided into the southwest region and Hangang River water system, and the AMOVA outcome among the groups was 80.27% (*F*c_T_ = 0.803), and among populations within the groups, it was 18.72% (*F*_SC_ = 0.949). In contrast, by dividing the groups based on haplotype distribution, the AMOVA showed a difference in variance, which among the groups was 98.41% (Fc_T_ = 0.984) and among the populations within groups was 0.42% (*F*_SC_ = 0.264).

## 4. Discussion

### 4.1. Genetic Diversity within 19 Populations of C. herzi

Genetic diversity is essential for the persistence and evolutionary potential of a species [21]. The genetic diversity in the *C. herzi* mtDNA was lower than in that of *Siniperca scherzeri* (*h*: 0.710–0.923, nucleotide diversity: 0.00099–0.00984 [22]) and *Siniperca chuatsi* (*h*: 0.8182–0.9091, nucleotide diversity: 0.0017–0.0031 [23]). Such low genetic diversity could be due to anthropogenic factors. Habitat destruction and overfishing are major factors that reduce genetic diversity [21,24]. *C. herzi* spawns under stones in shoals and gravel streams; therefore, the destruction of spawning grounds can lead to population decline and a reduction in genetic diversity [25,26]. The tasty food value of *C. herzi* may cause a reduction in its genetic diversity due to the possibility of overfishing. Additionally, the artificial breeding of *C. herzi* for the purpose of creating fishery resources may have caused a decrease in genetic diversity due to the inconsideration of the genetic factors of the mother used for breeding [27].

### 4.2. Genetic Structure of Wild and Translocated Populations

In this study, the genetic structure of *C. herzi* and the origin of the translocated population were investigated. The mtDNA datasets showed significant genetic differences in the Hangang River water system populations (HDC, HSC, HHC, HWS, HMW, and HSSC) and the YYND vs. Geumgang River populations (GYD and GND) vs. MJJ, SON, SJJ, and YJS vs. the Nakdonggang River populations (NNG, NDC, NSG, NDJ, NIM, and NYS). The haplotype was divided into four groups, and since the translocation population (the YYND population) shared the H1 haplotype with the Hangang River populations, this implied that the YYND population originated from the Hangang River populations. AMOVA supported the genetic differentiation of the four groups based on haplotype distribution. The AMOVA outcome of the Hangang River and southwest region water system populations was 80.27% among the groups, but that of the four groups according to haplotype distribution was 98.41%, which was genetically distinct. In addition, the ML tree supported the genetic differentiation pattern divided into four groups, and it was identical to the AMOVA and haplotype distribution. *C. herzi* was described by Park et al. [9], who suggested that the Hangang River water system populations were morphologically and embryologically different from other water system populations and were genetically different from the rest of the populations. Thus, in the future, conservation efforts will be required to identify the genetic structure of the population via the development of SNP markers with high resolution for *C. herzi*.

We describe the most likely scenario for *C. herzi* translocation in the YYND region based on the results of the genetic differences and haplotype sharing. The Hangang River population is geographically close to YYND and is prone to translocation by humans. In addition, since it was released to reinforce fishery resources, the YYND population may have originated from the Hangang River populations of the releasing mother population. Hypotheses strongly suggest that the YYND population originated from the Hangang River water system.

## 5. Conservation Implications

The Korean aucha perch is a commercially important species and a potentially valuable fishery resource. However, the aquaculture of *C. herzi* is not widely practiced, and wild populations are supplied primarily by fisheries. As a result, overfishing poses a threat to this species. They are currently released for the purpose of reinforcing fishery resources via aquaculture; however, owing to the lack of commercial systematic aquaculture technology, breeding, or genetic breeding systems, the individuals are obtained from wild individuals. Furthermore, threats such as human habitat destruction and overfishing are gradually reducing their population, which can lead to their extinction in the absence of active conservation efforts. In this study, the genetic diversity of *C. herzi* was lower than that of other freshwater fish and populations such as NNG and YYND. Given the genetic differences between populations, our analysis suggests that methods of population expansion via artificial breeding should be considered.

Conservation strategies should be planned according to the conservation management units. We found that in mtDNA, the Han River water population had a unique haplotype and genotype that was genetically distinct from that of the southwest water population. Hence, The Hangang River watershed population should be treated as a conservation unit separate from other southwestern watershed populations. However, the Geumgang River populations (GYD and GND) vs. MJJ, SON, SJJ, and YJS vs. the Nakdonggang River populations (NNG, NDC, NSG, NDJ, NIM, and NYS) should be managed by dividing the conservation units, because the southwestern water system is divided according to its unique genetic variation based on the mtDNA haplotype. Given that the translocated population (the YYND population) has been translocated to a previously uninhabited area, the negative ecological impact of the translocation should be assessed and considered for inclusion in a conservation management unit. Our results provide information on the genetic basis for the conservation of *C. herzi*.

## 6. Conclusions

The Korean endemic aucha perch, *Coreoperca herzi*, belongs to the family Centropomidae. Thus far, studies on *C. herzi* have focused on mitochondrial genomes, egg development, and early life history, while studies on their genetic diversity or genetic structure are lacking. We investigated these aspects in this study using mitochondrial DNA data. Haplotypes were divided into the Hangang River, Nakdonggang River, Geumgang River, and southwest region water system populations. A translocated population, the Yangyang Namdaechun Stream, was confirmed to have originated from the Hangang River water system population based on the haplotype distribution, genetic structure, and scenario results. According to the *COI* analyses, the analysis of molecular variance revealed a higher variance in the four water system groups (98.41%) than in the southwest region water system versus the Hangang River water system (80.27%) groups. This study presented basic data for conservation by providing extensive information on the genetic diversity, genetic structure, and translocation population of *C. herzi*.

## Figures and Tables

**Figure 1 animals-13-02614-f001:**
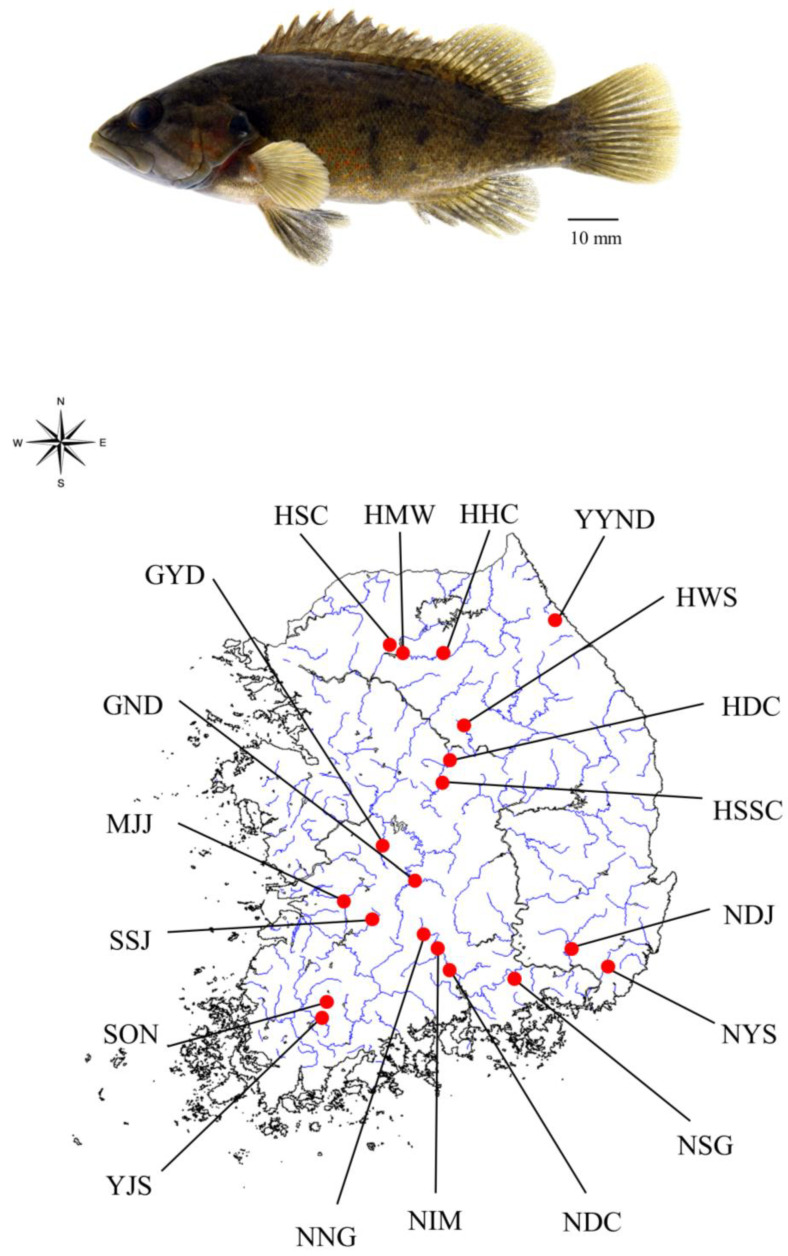
Specimen sample of *C. herzi* (**above**). A sample location of 19 populations of *C. herzi* (**below**).

**Figure 2 animals-13-02614-f002:**
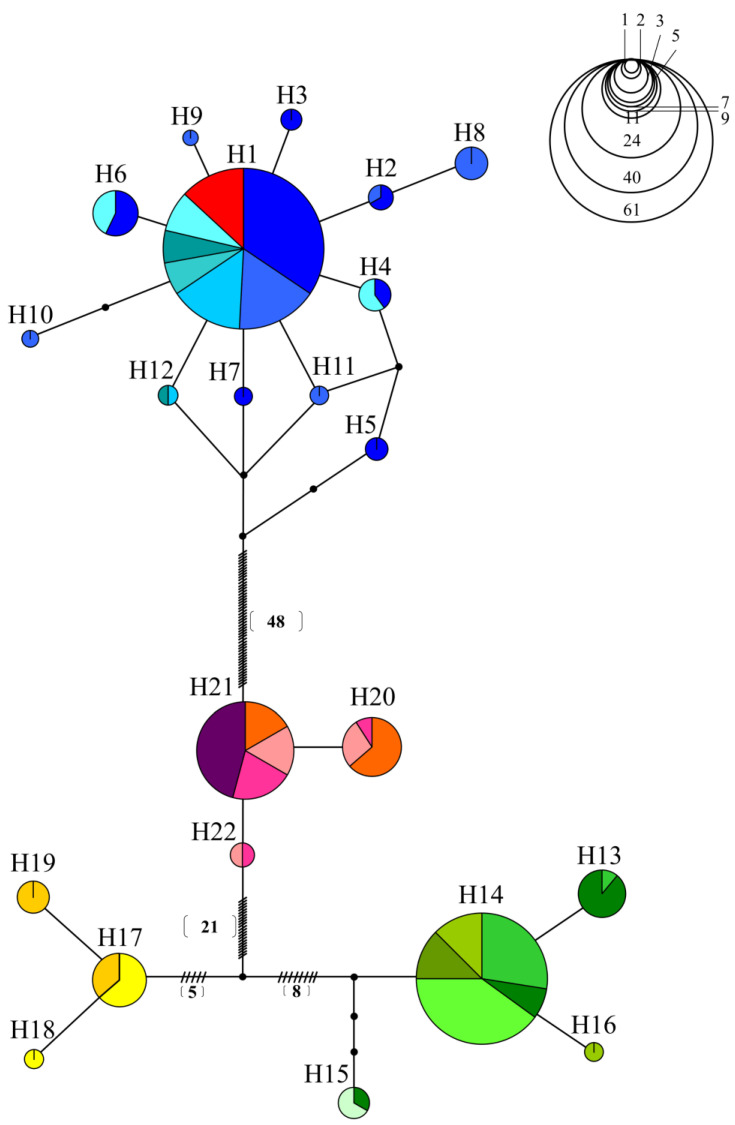
A haplotype network of 19 populations using mitochondrial *COI* genes from *C. herzi*.

**Figure 3 animals-13-02614-f003:**
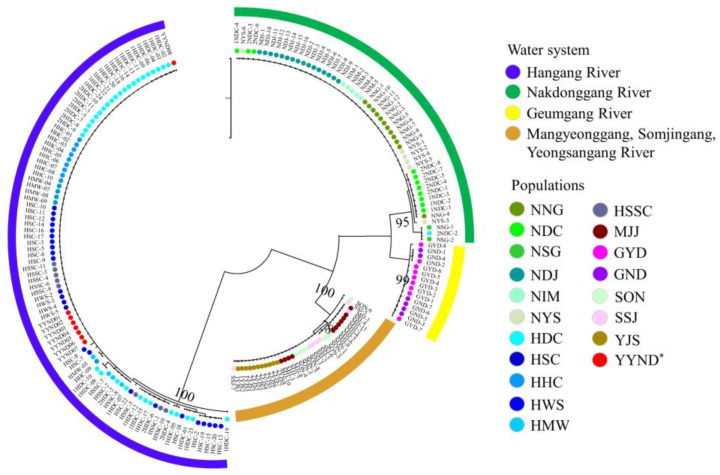
Population phylogenetic tree constructed using the PhyML program using the maximum likelihood (ML) method based on mtDNA sequences; *: indicates the introduced population.

**Table 1 animals-13-02614-t001:** Genetic diversity summary based on the mtDNA of *Coreoperca herzi*.

ID	MtDNA	*h*	*H_d_*	Nucleotide Diversity	*D*	*F*
NNG	12	2	0.16667	0.00025	−1.14053	−0.47566
NDC	12	3	0.53030	0.00176	−1.12253	0.97942
NSG	2	1	0.00000	0.00000	-	-
NDJ	16	1	0.00000	0.00000	-	-
NIM	5	1	0.00000	0.00000	-	-
NYS	6	2	0.33333	0.00051	−0.93302	−0.00275
HDC	34	7	0.60784	0.00144	−1.28905	−2.74768
HSC	19	6	0.67836	0.00196	−0.81109	−1.59902
HHC	10	2	0.20000	0.00031	−1.11173	−0.33931
HWS	4	1	0.00000	0.00000	-	-
HMW	5	2	0.40000	0.00061	−0.81650	0.09021
HSSC	11	3	0.70909	0.00133	0.85048	0.32272
MJJ	11	2	0.50909	0.00078	1.18556	1.02256
GYD	8	2	0.25000	0.00038	−1.05482	−0.18197
GND	7	2	0.57143	0.00087	1.34164	0.85642
SON	8	3	0.67857	0.00120	0.06935	−0.22360
SSJ	7	3	0.52381	0.00087	−1.23716	−0.92180
YJS	11	1	0.00000	0.00000	-	-
YYND	8	1	0.00000	0.00000	-	-

**Table 2 animals-13-02614-t002:** *F*_ST_ among 19 populations of *C. herzi* according to mtDNA datasets.

	NNG	NDC	NSG	NDJ	NIM	NYS	HDC	HSC	HHC	HWS	HMW	HSSC	MJJ	GYD	GND	SON	SSJ	YJS	YYND
NNG	-	0.322 **	0.963	0.025	−0.093	0.033	0.987 ***	0.985 ***	0.997 ***	0.998 ***	0.996 ***	0.991 ***	0.990 ***	0.988 ***	0.981 ***	0.987 ***	0.990 ***	0.997 ***	0.998
NDC	-	-	0.768 **	0.473 **	0.295	0.302	0.983 ***	0.979 ***	0.987 ***	0.984	0.983 ***	0.982 ***	0.975 ***	0.953	0.945 ***	0.970 ***	0.971 ***	0.982 ***	0.988 ***
NSG	-	-	-	1.000 ***	1.000 ***	0.934	0.984 ***	0.979 ***	0.997 ***	1.000	0.994	0.986 ***	0.987 ***	0.988	0.974 ***	0.980 ***	0.986 ***	1.000 ***	1.000 ***
NDJ	-	-	-	-	0.000	0.179	0.989 ***	0.988 ***	0.999 ***	1.000 ***	0.999 ***	0.994 ***	0.994 ***	0.995	0.990 ***	0.992 ***	0.995 ***	1.000 ***	1.000 ***
NIM	-	-	-	-	-	−0.034	0.985 ***	0.982 ***	0.998 ***	1.000 ***	0.996 ***	0.989 ***	0.989 ***	0.990	0.979 ***	0.985 ***	0.989 ***	1.000 ***	1.000 ***
NYS	-	-	-	-	-	-	0.985 ***	0.981 ***	0.996 ***	0.996 ***	0.994 ***	0.988 ***	0.986 ***	0.983	0.972 ***	0.982 ***	0.986 ***	0.997 ***	0.998 ***
HDC	-	-	-	-	-	-	-	0.113 *	0.002	−0.100	−0.022	0.008	0.984 ***	0.987	0.986 ***	0.983 ***	0.983 ***	0.986 ***	−0.023
HSC	-	-	-	-	-	--	-	-	0.114	0.012	0.065	0.178	0.981 ***	0.984 ***	0.983 ***	0.978 ***	0.979 ***	0.984 ***	0.095
HHC	-	-	-	-	-	-	-	-	-	−0.122	−0.126	0.160	0.993 ***	0.996 ***	0.994 ***	0.991 ***	0.993 ***	0.998 ***	−0.024
HWS	-	-	-	-	-	-	-	-	-	-	−0.053	0.049	0.993 ***	0.997 ***	0.994 ***	0.990 ***	0.993 ***	1.000 ***	0.000
HMW	-	-	-	-	-	-	-	-	-	-	-	0.099	0.991 ***	0.995 ***	0.992 ***	0.988 ***	0.990 ***	0.998 ***	0.101
HSSC	-	-	-	-	-	-	-	-	-	-	-	-	0.987 ***	0.990 ***	0.988 ***	0.984 ***	0.986 ***	0.992 ***	0.154
MJJ	-	-	-	-	-	-	-	-	-	-	-	-	-	0.986 ***	0.982 ***	0.026	0.275	0.600 **	0.994 ***
GYD	-	-	-	-	-	-	-	-	-	-	-	-	-	-	0.268	0.982 ***	0.986 ***	0.996	0.998 ***
GND	-	-	-	-	-	-	-	-	-	-	-	-	-	-	-	0.977 ***	0.981 ***	0.993 ***	0.996 ***
SON	-	-	-	-	-	-	-	-	-	-	-	-	-	-	-	-	−0.058	0.273 ***	0.993 ***
SSJ	-	-	-	-	-	-	-	-	-	-	-	-	-	-	-	-	-	0.069	0.995 ***
YJS	-	-	-	-	-	-	-	-	-	-	-	-	-	-	-	-	-	-	1.000 ***
YYND	-	-	-	-	-	-	-	-	-	-	-	-	-	-	-	-	-	-	-

*: Significant values (*p* < 0.05); **: significant values (*p* < 0.001); ***: significant values (*p* < 0.001).

**Table 3 animals-13-02614-t003:** Analysis of molecular variance (AMOVA) summary statistics for *C. herzi* populations.

Source of Variation	d.f.	Sum of Squares	Variance Components	Total Variance (%)	*F*-Statistics
mtDNA *COI* Southwest region water system vs. Hangang River water system (NNG, NDC, NSG, NDJ, NIM, NYS, MJJ, GYD, GND, SON, SSJ, YJS vs. HDC, HSC, HHC, HWS, HMW, HSSC, YYND)					
Among groups	1	2319.321	22.92867	80.27	*F*c_T_ = 0.803 ***
Among populations within groups	17	889.882	5.34772	18.72	*F*_SC_ = 0.949 ***
Within populations	177	50.884	0.28748	1.01	*F*_ST_ = 0.990 ***
Total	195	3260.087	28.56387	100.0	
mtDNA *COI* (NNG, NDC, NSG, NDJ, NIM, NYS vs. GYD, GND vs. MJJ, SON, SSJ, YJS vs. HDC, HSC, HHC, HWS, HMW, HSSC, YYND)					
Among groups	3	3189.688	24.26246	98.41	*F*c_T_ = 0.984 ***
Among populations within groups	15	19.515	0.10333	0.42	*F*_SC_ = 0.264 ***
Within populations	177	50.884	0.28748	1.17	*F*_ST_ = 0.988 ***
Total	195	3260.087	24.65327	100.0	

d.f.: Degrees of freedom; *** *p* < 0.001. *F*_ST_ is based on standard permutation across the full dataset.

## Data Availability

All genotypes and related information are available upon reasonable request (e-mail: kimkangrae9586@gmail.com).

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
