# Peer review of "Genetic Structure and Genetic Diversity of the Endemic Korean Aucha Perch, Coreoperca herzi (Centropomidae), in Korea"

_animals, 2023, doi:10.3390/ani13162614_

Round 1
Reviewer 1 Report
The authors analyzed the genetic structure and genetic diversity of the endemic Korean Aucha Perch. The topic is interesting and falls within th escope of Animals. The manuscript is well written and very nicely presented. I have only one concern which is the number of microsatellites used by the authors. I do believe that the 7 marker is simply not ennough, besides, the genetic diversity was between 2.3 and 7.3. As far as I know a rule of thumb is that it should be at least 4. In my opnion it would be better if authors would remove the microsatellite part altogether making the article more coherent. This is why I suggested major revision.
Author Response
Response author
Q1.
The authors analyzed the genetic structure and genetic diversity of the endemic Korean Aucha Perch. The topic is interesting and falls within th escope of Animals. The manuscript is well written and very nicely presented. I have only one concern which is the number of microsatellites used by the authors. I do believe that the 7 marker is simply not ennough, besides, the genetic diversity was between 2.3 and 7.3. As far as I know a rule of thumb is that it should be at least 4. In my opnion it would be better if authors would remove the microsatellite part altogether making the article more coherent. This is why I suggested major revision.
A1. First of all thanks for the review. According to the reviewer's opinion, the MS marker part was deleted.

Reviewer 2 Report
This is the first study considering the genetic diversity of Coreoperca herzi, which belongs to the family Centropomidae, endemic from Corea. To this end, they collected samples from 19 spots and were able to find great genetic diversity (halotypes). Thus, this study has potential impacts on the strategies of conservation. The authors used microsatellite and mitochondrial DNA data and they were able to develop 7 microsatellite markers that satisfied the condition of having a polymorphism information content value of ≥ 0.5 and with no deviation from the Hardy–Weinberg equilibrium out of 100 markers.
The manuscript is interesting, and well-written, the data is convincing and the experiments and methods appropriate. I just missed an illustration of the organism and a map of the sample´s location. I think those details will enrich the article, making it more appealing to a broad range of readers.
Author Response
Q1.
This is the first study considering the genetic diversity of Coreoperca herzi, which belongs to the family Centropomidae, endemic from Corea. To this end, they collected samples from 19 spots and were able to find great genetic diversity (halotypes). Thus, this study has potential impacts on the strategies of conservation. The authors used microsatellite and mitochondrial DNA data and they were able to develop 7 microsatellite markers that satisfied the condition of having a polymorphism information content value of ≥ 0.5 and with no deviation from the Hardy–Weinberg equilibrium out of 100 markers.
The manuscript is interesting, and well-written, the data is convincing and the experiments and methods appropriate. I just missed an illustration of the organism and a map of the sample´s location. I think those details will enrich the article, making it more appealing to a broad range of readers.
A1.
Thanks for the review. Edited based on reviewer comments.

Reviewer 3 Report
The manuscript "Genetic Structure and Genetic Diversity of the Endemic Korean Aucha Perch, Coreoperca herzi (Centropomidae), in Korea" by Kim et al., aims to describe the genetic structure and diversity of the korean aucha perch in four distinct geographical aquatic areas. Moreover, they analysed the demographic history of a single population.
It is an interesting study, however methods need to be more explicit.
My major concern is the ABC approach. The authors should discuss in the Introduction in more detail about the potential environmental pressures, and state more clear hypotheses regarding the demography of the species. Relying on the latter, they should describe and explain the ABC models they tend to test.
Also, authors should check and analyse for Linkage disequilibrium on the 7 tested microsatellites.
Thus, I suggest major revision of the ms prior publication.
Some more specific comments below
line 64: How did you obtain the fishes? Did you fish them, or did you purchase them from local markets? The authors should define this, since there is a possibility that local markets may sell products from different regions.
line 70: How many samples did you use for the assembly?
line 79: Why did you choose randomly?
line 99: Since you referred before the quantity of the DNA (50ng), you should also refer to it here as well.
line 109-112: HWE should go before the calculations of genetic indices. Also, linkage disequilibrium analysis should be performed prior genetic diversity calculations. LD could be assessed through genepop as well.
line 113: Authors should mention the alignment software and the algorithm used.
line 119-120: Authors should mention the groups of populations for the AMOVA.
line 122-123: Since you have 19 populations, why did you used K 1-10?
line 130: replace "statistical" with "demographic".
line 131: Authors should describe here in M&Ms the scenarios, the parameters and the prior range they used. Also, they have to describe the analysis they followed for the model selection.
Results
Authors should mention the Microchecker results, and the LD analysis before anything else. It is essential for microsatellite studies of null alleles and LD.
Table1: Authors should describe it in the legend. (e.g. Ho (observed heterozygosity etc). Also, it is not clear if the genetic indices are referred to the microsatellites, or to the MtDNA. Please be explicit.
line 160-162: This is discussion. Please move it to the relevant section.
Table 2: Authors should mention that above the diagonal are the Φst and below diagonal the Fst values. Also, you should describe in the M&Ms the test you did for the MtDNA Φst. Also, bold values are not very clear. I suggest authors use asterisks to denote significance.
line 186-188: This is discussion. Move it to the relevant section or delete it please.
line 197-198: This is discussion. Move it to the relevant section or delete it please.
Table 3: It could be moved to the Supplementary.
Figure4: principal components percentages should be referred on the axes.
Table4: There are tests with relatively low variance (e.g. within populations - Southwest region water system vs Hangang River water system), yet they are significant. Why do you believe this is happening?
line 269-276: This is the first time that scenarios are presented in the text. Authors should describe and explain why they used these three scenarios. I don't see any hypotheses stating why to draw such demography. Also, prior ranges and posterior values are missing.
line 299-303: There is no such information with relative references in the Introduction about the fishing pressure of the area. Please add relative text to the Introduction to support this in Discussion.
line 307-310: This is speculation. Please expand it with relative references, or delete it.
line 315-316: How do you know that these populations are vulnerable to extinction? Inbreeding doesn't always imply extinction.
line: 320: Authors mentioned that C. herzi suffered from bottlenecks. References are missing. Did you perform any bottleneck analysis?
line 320-321: Authors say that the YYND pop requires specific efforts to increase genetic diversity. What are these efforts? The authors should discuss this, presenting potential measurements and conservation efforts.
line 356-368: This is not well-described at all. Since there is no adequate explanation in the M&Ms, authors speculate of the "correct" model. Parameters estimated values are not discussed, and are not presented in the Results either.
Minor editing
Author Response
Response author
The manuscript "Genetic Structure and Genetic Diversity of the Endemic Korean Aucha Perch, Coreoperca herzi (Centropomidae), in Korea" by Kim et al., aims to describe the genetic structure and diversity of the korean aucha perch in four distinct geographical aquatic areas. Moreover, they analysed the demographic history of a single population.
It is an interesting study, however methods need to be more explicit.
My major concern is the ABC approach. The authors should discuss in the Introduction in more detail about the potential environmental pressures, and state more clear hypotheses regarding the demography of the species. Relying on the latter, they should describe and explain the ABC models they tend to test.
Also, authors should check and analyse for Linkage disequilibrium on the 7 tested microsatellites.
Thus, I suggest major revision of the ms prior publication.
Q1.
line 64: How did you obtain the fishes? Did you fish them, or did you purchase them from local markets? The authors should define this, since there is a possibility that local markets may sell products from different regions.
A1. Thanks for the review. I collected them using a fishing net.
Q2.
line 70: How many samples did you use for the assembly?
line 79: Why did you choose randomly?
line 99: Since you referred before the quantity of the DNA (50ng), you should also refer to it here as well.
line 109-112: HWE should go before the calculations of genetic indices. Also, linkage disequilibrium analysis should be performed prior genetic diversity calculations. LD could be assessed through genepop as well.
A2. Thanks for the review. Please understand that the MS part has been deleted according to the opinion of the first reviewer, so we cannot respond to your opinion.
Q3.
line 113: Authors should mention the alignment software and the algorithm used.
A3. Thanks for the review. Edited according to comments.
[mtDNA was examined through alignment using ClustalW algorithm of the MEGA software [16] based on COI sequencing.]
Q4.
line 119-120: Authors should mention the groups of populations for the AMOVA.
A1. Thanks for the review. Edited according to comments.
[The first, AMOVA was divided according to the distribution by water system (Southwest region water system vs Hangang River water system; NNG, NDC, NSG, NDJ, NIM, NYS, MJJ, GYD, GND, SON, SSJ, YJS vs HDC, HSC, HHC, HWS, HMW, HSSC, YYND). Second, AMOVA was divided according to the haplotype distribution (NNG, NDC, NSG, NDJ, NIM, NYS vs GYD, GND vs MJJ, SON, SSJ, YJS vs HDC, HSC, HHC, HWS, HMW, HSSC, YYND).]
Q5.
line 122-123: Since you have 19 populations, why did you used K 1-10?
line 130: replace "statistical" with "demographic".
line 131: Authors should describe here in M&Ms the scenarios, the parameters and the prior range they used. Also, they have to describe the analysis they followed for the model selection.
A5. Thanks for the review. Please understand that the MS part has been deleted according to the opinion of the first reviewer, so we cannot respond to your opinion.
Q6.
Results
Authors should mention the Microchecker results, and the LD analysis before anything else. It is essential for microsatellite studies of null alleles and LD.
Table1: Authors should describe it in the legend. (e.g. Ho (observed heterozygosity etc). Also, it is not clear if the genetic indices are referred to the microsatellites, or to the MtDNA. Please be explicit.
A6.
Thanks for the review. Please understand that the MS part has been deleted according to the opinion of the first reviewer, so we cannot respond to your opinion.
Q7.
line 160-162: This is discussion. Please move it to the relevant section.
A7. Thanks for the review. Please understand that the MS part has been deleted according to the opinion of the first reviewer, so we cannot respond to your opinion.
Q8.
Table 2: Authors should mention that above the diagonal are the Φst and below diagonal the Fst values. Also, you should describe in the M&Ms the test you did for the MtDNA Φst. Also, bold values are not very clear. I suggest authors use asterisks to denote significance.
A8. Thanks for the review. Please understand that the MS part has been deleted according to the opinion of the first reviewer, so we cannot respond to your opinion.
Q9.
line 186-188: This is discussion. Move it to the relevant section or delete it please.
line 197-198: This is discussion. Move it to the relevant section or delete it please.
A9. Thanks for the review. Please understand that the MS part has been deleted according to the opinion of the first reviewer, so we cannot respond to your opinion.
Q10.
Table 3: It could be moved to the Supplementary.
A10.
Thanks for the review. The table has been moved to supplementary material.
Q11.
Figure4: principal components percentages should be referred on the axes.
A11.
Thanks for the review. Please understand that the MS part has been deleted according to the opinion of the first reviewer, so we cannot respond to your opinion.
Q12.
Table4: There are tests with relatively low variance (e.g. within populations - Southwest region water system vs Hangang River water system), yet they are significant. Why do you believe this is happening?
A11.
Thanks for the review. According to Part et al., 2020, it is judged that a significant result has been obtained because the Hangang River water system has a different pattern of color and the possibility of other species cannot be ruled out. Since a follow-up study on this is in progress, the content has not been added in this paper.
[Park, J.M.; Jeon, H.B.; Suk, H.Y.; Cho, S.J.; Han, K.H. Early life history of Coreoperca herzi in Han River, Korea. Dev Reprod. 2020, 24, 63–70. DOI:10.12717/DR.2020.24.1.63.]
Q13.
line 269-276: This is the first time that scenarios are presented in the text. Authors should describe and explain why they used these three scenarios. I don't see any hypotheses stating why to draw such demography. Also, prior ranges and posterior values are missing.
A13.
line 299-303: There is no such information with relative references in the Introduction about the fishing pressure of the area. Please add relative text to the Introduction to support this in Discussion.
A13.
line 307-310: This is speculation. Please expand it with relative references, or delete it.
line 315-316: How do you know that these populations are vulnerable to extinction? Inbreeding doesn't always imply extinction.
line: 320: Authors mentioned that C. herzi suffered from bottlenecks. References are missing. Did you perform any bottleneck analysis?
line 356-368: This is not well-described at all. Since there is no adequate explanation in the M&Ms, authors speculate of the "correct" model. Parameters estimated values are not discussed, and are not presented in the Results either.
A13.
Thanks for the review. Please understand that the MS part has been deleted according to the opinion of the first reviewer, so we cannot respond to your opinion.
Q14.
line 320-321: Authors say that the YYND pop requires specific efforts to increase genetic diversity. What are these efforts? The authors should discuss this, presenting potential measurements and conservation efforts.
A14.
Thanks for the review. One of the efforts is to augment population health. One way to reinforce these population fitness is through population release through artificial breeding. This was mentioned in the manuscript. Also, in the case of YYND, it is a translocated population, and is now in a place where it did not exist originally. Therefore, after a negative evaluation of this, it was mentioned in the manuscript that conservation efforts are needed.
[Given the genetic differences between populations, our analysis suggests that methods of population expansion through artificial breeding be considered.]
[Given that the translocated population (the YYND population) has been translocated to a previously uninhabited ar-ea, the negative ecological impact of the translocation should be assessed and considered for inclusion in a conservation management unit.]

Round 2
Reviewer 1 Report
I am pleased to acknowledge that the MS part has been removed. I have nor further problem with the article.
Reviewer 3 Report
Thank you for the responses. However, it's quite a pity that you took out the microsatelite analysis. You could fight back the other reviewer, but the final decision it's up to you.
Congrats on the new version of the ms.
Minor editing.